# Unilateral and Bilateral Strength Asymmetry among Young Elite Athletes of Various Sports

**DOI:** 10.3390/medicina56120683

**Published:** 2020-12-10

**Authors:** Maros Kalata, Tomas Maly, Mikulas Hank, Jakub Michalek, David Bujnovsky, Egon Kunzmann, Frantisek Zahalka

**Affiliations:** Sport Research Center, Faculty of Physical Education and Sport, Charles University, Jose Martiho 31, 16252 Prague 6, Czech Republic; kalata@ftvs.cuni.cz (M.K.); hank@ftvs.cuni.cz (M.H.); jak.michalek@seznam.cz (J.M.); bujnovsky@ftvs.cuni.cz (D.B.); egonkunzmann@gmail.com (E.K.); zahalka@ftvs.cuni.cz (F.Z.)

**Keywords:** muscular symmetry, isokinetic peak torque, maladaptation, youth, lower limbs, soccer

## Abstract

*Background and objective:* Type of physical activity may influence morphological and muscular asymmetries in the young population. However, less is known about the size of this effect when comparing various sports. The aim of this study was to identify the degree of bilateral asymmetry (BA) and the level of unilateral ratio (UR) between isokinetic strength of knee extensors (KE) and flexors (KF) among athletes of three different types of predominant locomotion in various sports (symmetric, asymmetric and hybrid). *Material and methods:* The analyzed group consisted of young elite athletes (*n* = 50). The maximum peak muscle torque of the KE and KF in both the dominant (DL) and non-dominant (NL) lower limb during concentric muscle contraction at an angular velocity of 60°·s^−1^ was measured with an isokinetic dynamometer. *Results:* Data analysis showed a significant effect of the main factor (the type of sport) on the level of monitored variables (*p* = 0.004). The type of sport revealed a significant difference in the bilateral ratio (*p* = 0.01). The group of symmetric and hybrid sports achieved lower values (*p* = 0.01) of BA in their lower limb muscles than those who played asymmetric sports. The hybrid sports group achieved higher UR values (*p* = 0.01) in both lower limbs. *Conclusions:* The results indicate that sports with predominantly symmetrical, asymmetrical, and hybrid types of locomotion affected the size of the BA, as well as the UR between KE and KF in both legs in young athletes. We recommend paying attention to regular KE and KF strength diagnostics in young athletes and optimizing individual compensatory exercises if a higher ratio of strength asymmetry is discovered.

## 1. Introduction

Various movement and sports activities differ not only in specific physical tasks but also in the volume and intensity of the physical load on the human body and its predominant character, thought of as a differentiating criterion that assesses the occurrence and predominance of cyclic and acyclic movement patterns during sports performance. The highest performance levels of in athletic disciplines are associated with the targeted development of sport-specific anthropometric, motor, and physiological characteristics [1]. Hart et al. [2] stated that morphological asymmetries are an adaptive consequence, amplified by long-term and intensive activities in the selected sports specialization as intense, year-round training in one sport at the exclusion of other sports [3]. Long-term preferred and uncompensated loads on one side of the body may lead to asymmetry and dominance of one leg, which can be a result of pre-existing limb preference (footedness, handedness). When this situation is repeated over the many years in which an athlete practices repetitive asymmetric loading, health problems such as lower back pain can occur [4]. Even if training is conducted correctly, the specifics of the physical stresses imposed by the discipline could be strong enough for the athletes to inevitably develop a certain degree of functional and morphological asymmetry as adaptive changes occur in the dominant side [5]. It was reported that the risk of sustaining tissue damage increases in situations when the load exerted on a given tissue exceeds its tolerance, for example, in unilateral activities [6]. For example, it has been reported that limb preference in soccer may lead to strength asymmetries, which may result in large changes in the myo-dynamic characteristics of the muscle, particularly in the dominant leg [7]. Athletes who undergo specific training from an early age and dominantly focus on one type of physical activity (early specialization) can be deprived of year-round general motor learning from various activities and thus miss attainment of certain fundamental motor skills [8]. Several studies reported [9,10], that values of morphological or muscle strength (MS) asymmetry varying by 10% to 15% are usually considered starting indicators of significant unbalance and are associated with a higher risk of injury and lower specific performance due to an imbalance in the manifestation of bilateral and unilateral MS. One of the main methods for identifying and comparing MS asymmetries is the use of isokinetic dynamometry [11]. With proper evaluation, it is possible to influence or correctly develop an individual training process for balanced strength development and, with normative values, it is possible to better understand differences in terms of specific sports and gender. For example, in addition to running, kicking is a basic soccer skill. It requires asymmetric movement patterns and predominantly loads the lower limbs [12]. Muscle imbalances in elite soccer players influenced specific skills (e.g., kicking accuracy) negatively [12].

Sports can be classified from various points of view, but the scientific field lacks a categorization that would include the predominant character of physical activity due to its frequency during sports performance. Cyclic sports predominantly involve periodically repeated movements such as medium to long-distance running, which may hypothetically lead to a balanced unilateral load on the body with low inter-limb strength asymmetry [13,14,15]. Conversely, Meyers et al. [16] reported that running biomechanics are generally symmetric; however, some kinetic asymmetry has been detected in healthy young boys. On the other hand, an acyclic character (unilateral load of one or more body segments) can be found in sports such as tennis (swings with a tennis racket), volleyball (spikes), or fencing. In other sports such as soccer, handball, and volleyball, there are hybrid actions with intermittent and irregularly alternated cyclic and acyclic movements. If one side of the body dominates over the other side because of the requirements of the specific movement (e.g., in ice-hockey, hitting, pushing, and semi-crouched position with side bending and trunk rotation), this generates additional loads on the body and could influence lower back problems [17] Obviously, the degree of a particular type of movement repetition varies from sport to sport, and thus affects the extent of possible asymmetry. Untreated asymmetry may lead to unbalanced changes in the volume and condition of muscle tissue and its shortening, which limits the range of joint mobility. Thus, monitoring and regulating of MS unbalance should lead to performance improvement and lowering of the risk of injury [18]. Therefore, it is very important to examine and clarify the extent to which these sports differ in terms of the natural shaping of asymmetry.

Coratella et al. [19] monitored elite soccer players under 21 years old and found that a higher unilateral ratio (UR) of MS between hamstring muscles (KF—knee flexors) and quadriceps muscles (KE—knee extensors) showed a positive relationship with performance in change of direction speed (COD) and linear sprint. Maly et al. [20] stated that the preference for one type of physical activity influences morphological and muscular asymmetries in the population under 15 years old. Bilateral asymmetry (BA) differences in the MS of KE were significantly higher compared to the inactive population. On the other hand, these differences were insignificant compared to floorball players. Elite adult soccer players cover a total distance covered of up to 10–12 km during a match [21]. In contrast, handball players achieve a total covered distance in a match of only between 4–5 km [22], however, they perform intensive movements (like jumps) more frequently than soccer players (up to 90 per game) [23]. American football players with greater than three years of training experience showed significantly larger morphological asymmetries between their limbs (bone area, areal bone mineral content, and areal bone mineral density) than players with less than three years of training experience [2]. Schleichartdt et al. [24] reported in elite youth volleyball players significantly stronger internal rotators of the dominant shoulders compared to the non-dominant side. Obviously, besides other factors such as individual technical skills, various MS of the lower limbs can have different effects on performance. Morouco et al. [25] reported deficits in upper limb MS in young swimmers (15.6 ± 2.1 years). Up to 66.7% of swimmers showed asymmetric strength in the first ten strokes, that is asymmetry index (peak forces DL vs. NL) higher than 10%. However, higher MS asymmetry did not lead to lower performance. Luk et al. [26] reported a lower level of asymmetry in the MS that is produced by the lower limbs in powerlifters during a unilateral jumping test (2.74 ± 0.74%) compared to that produced by high jumpers (6.73 ± 1.84%).

The aim of this study was to identify the degree of bilateral asymmetry and the level of unilateral ratio between isokinetic muscle strength of knee extensors and flexors in athletes, taking into account different types of predominant locomotion (symmetric, asymmetric, and hybrid). We assume a significantly (*p* < 0.05) higher degree of lower limb muscle strength asymmetries in athletes who predominantly perform acyclic movements.

## 2. Materials and Methods

### 2.1. Subject

The analyzed group consisted of young elite male (*n* = 31, age = 14.6 ± 1.5 years) and female (*n* = 19, age = 14.2 ± 1.1 years) athletes. A minimum of five years of specialization in one sport was a criterion for inclusion in this study. Sports specialization is understood as regular, intense, year-round training primarily in one sport to the exclusion of other sports [3]. The athletes were assigned to three groups according to their sports specialization and predominant physical load. First group of “symmetrical” sports (SY)—Predominance of cyclic movements (triathlon and sport aerobics), (*n* = 13, age = 15.25 ± 0.94 years; body weight = 53.69 ± 8.90 kg; body height = 168.23 ± 9.50 cm). Second group of “asymmetrical” sports (ASY)—Predominance of acyclic movements (tennis and volleyball), (*n* = 22, age = 14.51 ± 1.95 years; body weight = 56.55 ± 13.53 kg; body height = 163.70 ± 16.43 cm). Third group of “hybrid” sports (HY)—Predominance of combined movements (soccer), (*n* = 15, age = 15.56 ± 0.46 years; body weight = 67.33 ± 5.94 kg; body height = 173.2 ± 5.36 cm). The research was conducted using non-invasive methods according to the ethical standards of the Ethical Committee of the Faculty of Physical Education and Sport, Charles University, in Prague, Czech Republic (code: 238/2019; 7 November 2019). The ethical documents were prepared following the ethical standards of the Declaration of Helsinki and according to other ethical standards in sport and exercise science research [27].

### 2.2. Procedures

Evaluation of bilateral and ipsilateral strength ratio.

Lower limb MS was measured with an isokinetic dynamometer (Cybex NORM^®^, Humac, CA, USA). To ensure that relatively young players without prior experience of isokinetic measurement reach their maximum values, a slow angular velocity at 60°·s^−1^ was chosen to evaluate the maximum peak muscle torque of the KE and KF in both DL and NL during concentric muscle contraction [12]. We observed BA between the DL and NL (H:H—hamstring to hamstring; Q:Q—quadriceps to quadriceps) and unilateral strength asymmetries between the KF and KE (H:Q—hamstring to quadriceps). In our study, musculoskeletal abnormality of knee muscles was defined as a bilateral strength imbalance of more than 10% [28] which contributes to a knee risk factor [7]. Limb dominance was operationally defined as the foot the participant preferred to use to kick the ball. The tested subject sat on an ergonomically set dynamometer seat, with the arm of the dynamometer adjusted according to the instructions and the individual somatic characteristics of the participant. The axis of the dynamometer arm rotation was visually adjusted to the axis of knee rotation with a laser pointer. Peak torque was controlled by gravity correction. The motion range was 90° (maximum extension was marked and set as anatomic zero “0°”). The participant’s trunk and the thigh of the tested leg were fixed by straps because of the isolation of the examined movement. The participant held the side handles of the device during the measurement. Before the measurement, all subjects completed a standardized warm-up mainly focused on the quadriceps and hamstring muscle groups (5 min of indoor cycling and 2 sets with 10 repetitions of front squats, front lunges, and glute bridges). The test protocol consisted of five sub-maximal trial concentric repetitions for KF and KE. Between the individual sets, a standard rest interval (20 s) was included. Subsequently, two concentric attempts with maximum effort were performed. Verbal and visual feedback was provided throughout the tests. For further processing, the better result from the two attempts was selected.

### 2.3. Statistical Analysis

For the descriptive processing of the research data, arithmetic means and standard deviations were used. To verify the normality of the data distribution, the Shapiro-Wilk test was used, while homogeneity of variance was tested with Levene’s test. To determine the effect of the monitored factor (independent variable: type of sport) on differences in the observed maladaptive indicators (dependent variables), multiple analysis of variance (MANOVA) was used. Subsequently, Bonferroni’s post hoc analysis was used for multiple comparisons of the differences in the means of the individual groups. The effect size was determined using partial Eta square “$\eta_{p}^{2}$”. The differences in the monitored parameters were considered significant when *p* < 0.05. Statistical analysis was performed using IBM SPSS^®^ software (IBM Statistical Package for Social Science^®^ v. 21, Chicago, IL, USA).

## 3. Results

A significant effect of the main factor (type of sport) on the level of monitored variables (F_8,88_ = 3.09, *p* = 0.004, λ = 0.610) was found. The type of sport revealed significant difference in the bilateral ratio (F_2,47_ = 4.74, *p* = 0.01, $\eta_{p}^{2}=0.17$). Significantly higher values of BA in KE (Q:Q) were found among the SY sports group (5.42 ± 3.02%) and the ASY sports group (9.40 ± 4.32%). Athletes who performed typically asymmetrical physical activities also achieved significantly higher values than those who performed hybrid sports (5.46 ± 5.7%). Similarly, when comparing the strength asymmetries in the KF, the highest values were found in the ASY group (11.74 ± 7.41%) and the lowest values were in the HY group (6.97 ± 7.01%). However, the type of sport as main factor was not significantly different among groups (F_2,47_ = 2.08, *p* = 0.14, $\eta_{p}^{2}=0.08$). The UR of the DL and NL showed a significant dependence on the type of sport (Table 1). The post hoc analysis revealed a significantly higher H:Q ratio in the group of hybrid sports compared to group of asymmetric sports (HY:HQ_DL_ = 59.66 ± 8.81% vs. ASY:HQ_DL_ = 51.18 ± 6.57%, *p* < 0.05, i.e., HY:HQ_NL_ = 60.41 ± 9.87% vs. ASY:HQ_NL_ = 51.81 ± 7.98%, *p* < 0.05).

## 4. Discussion

### 4.1. Bilateral Strength Asymmetries

The results revealed a significant effect of type of sport on the level of bilateral muscular strength asymmetry in knee extensors. ASY group achieved significantly higher values of BA (Q:Q = 9.40 ± 4.32%) than athletes from the HY group (Q:Q = 5.46 ± 5.7%) and the SY group (Q:Q = 5.42 ± 3.02%). Our results support the hypotheses that specific activities required in sports disciplines affect the level of BA [29].

In volleyball, there are numerous repetitions of vertical jumps from the takeoff leg. Landing takes place in a strongly asymmetric position and causes large eccentric involvement of the lower limb muscles [23]. Depending on their playing position, volleyball players make between 300 and 450 vertical jumps and spikes during a training session [30]. Hadzic et al. [31] also reported higher average values of BA in the extensors (Q:Q = 12 ± 12.3%) in different age categories of elite volleyball players. Volleyball players in our study represented the ASY group and also reached the highest values of asymmetry (11.74 ± 7.41%) among the SY or HY group. Furthermore, tennis players were also included in the ASY group. Tennis is characterized by a series of dynamic actions, changes in directions, and rapid accelerations/decelerations. These can result in functional asymmetries of the lower limbs in young and adult tennis players [32]. Although the SY group achieved the lowest BA (5.42 ± 3.02), it is necessary to mention the fact that BA was also found in symmetrical sports such as long-distance runners [33]. In this group of athletes it has been reported that lateral asymmetries in KE may be linked to pronounced lateral asymmetries in tibial load during running. Moreover, lateral asymmetries detected during running using the triaxial inertial sensors might also reflect a sensible and effective measure for the runner to compensate for individual anatomic and/or orthopaedical conditions, such as scoliosis or a congenital/acquired articular malalignment [34]. A study on cycling (SY) reported increased asymmetries in the movements of the dominant (DL) and non-dominant (NL) lower limbs (kinematics and peak crank torques) with increasing load or fatigue on the DL [35].

The HY group in our study (young elite soccer players), showed lower values of BA (Q:Q) compared to the group included in the study of Maly et al. [20] (5.46 ± 5.7% vs. 10.99 ± 7.75%). In contrast, Silva et al. [36] presented closer values of BA in KEs (Q:Q = 6.28 ± 4.79%) in elite soccer players, although the study tested athletes aged between 18 and 20 years old. The disparity between the results could have been caused by a variant duration of sports specialization, different performance levels, specific volume of strength training, and compensation programs. Therefore, we must consider regular diagnostics of each team even within the same type of sport, due to the individual approach to training strategy. Magalhaes et al. [37] compared BA in KEs between adult soccer players and volleyball players and a greater degree of asymmetry was found in the volleyball group (10.1 ± 6.9% vs. 7.3 ± 6.5%). Similar to our study, the ASY group with the volleyball players reached a greater KE and KF asymmetry, but concerning strength asymmetries in KFs, the type of sport did not show any significant difference between the compared groups (*p* > 0.05). Higher values were found in the ASY group (11.74 ± 7.41%) than in the HY group (6.97 ± 7.01%) and the SY group (8.3 ± 7.79%). Bonetti et al. [38] also presented BA in KFs (H:H = 10.35%) in young soccer players. Interestingly, the deficit decreased with higher angular velocities (60°·s^−1^ vs. 240°·s^−1^, 10.35% vs. 4.79%), which disagrees with the study of Maly et al. [39] where no changes in BA within higher velocities were found. This represents the inter-individual differences among diverse teams of the same sport due to different criteria such as performance level, age of experiences, different strength programs, and training philosophy. Our study revealed similar bilateral differences in KEs (Q:Q = 5.42 vs. 9.40 vs. 5.46%) and KFs (H:H = 8.30 vs. 11.74 vs. 6.97%) in comparison to data for KEs (8.70 ± 6.89%) and KFs (15.62 ± 8.75%) in young athletes (soccer, rugby, basketball, and field hockey) reported by Locki et al. [40]. On the contrary, Maly et al. [20] reported an opposite ratio for floorball players (H:H = 7.73 ± 5.18%, Q:Q = 9.18 ± 8.01%) and a very balanced ratio for soccer players (H:H =10.86 ± 8.44%, Q:Q = 10.99 ± 7.75%). Vargas et al. [10] stated that BA was stable for all groups of various aged female soccer players, except for the under 13 years category (Q:Q = 11.2 ± 16.2%), where the BA of KEs exceeded 10%. The results of our study showed a higher standard deviation for BA in KFs (7.01 to 7.79%) compared to KEs (3.02 to 5.7%). This result emphasizes the inter-individual characteristics and supports the need to evaluate each athlete individually and to detect his weak points. In terms of the soccer players, Iga et al. [41] reported that players rarely use both legs with equal emphasis, because their preference to use one side more than the other is related to hemispheric dominance of the brain in the opposite site. This fact can influence some preferences, which can result in different morphological and pathophysiological maladaptation of athletes where one side is preferred compare to the other.

### 4.2. Unilateral Ratio between Knee Flexors and Extensors

This study showed a significant effect of the type of sport on the size of the UR for the DL and NL (Table 1). A significantly higher (*p* < 0.05) H:Q ratio for DL was found in the HY group (HQ_DL_ = 59.66 ± 8.81%) compared to the ASY group (HQ_DL_ = 51.18 ± 6.57%) and for NL was also found in the HY group (HQ_NL_ = 60.41 ± 9.87%) compared to the ASY group (HQ_NL_ = 51.81 ± 7.98%). The conventional ratio of KEs and KFs ranges between 55%–77% and the lower recommended limit increases by up to 60% [42]. The mean value of 55% was not reached by the ASY group, regardless of limb dominance. One of the reasons for this may be the fact that great emphasis in the training process is placed on the development of the MS of the KEs, which are primarily responsible for jumping in volleyball. Meanwhile, adequate training of their antagonists (KFs) is often forgotten, despite them playing a primary role in landing strategies [23].

Dalton et al. [43] observed a higher frequency of hamstring injuries in field sports (soccer and hockey) than in court sports (basketball, volleyball). Ekstrand et al. [44] reported that most injuries are from non-contact mechanisms, with the most common mechanisms being running and sprinting activities occurring during sports, which again emphasizes the athlete’s adequate MS level. Male elite soccer players did not meet the reference values (>60%) used to assess the balance between MS of KFs and KEs [45]. This is the reason why early diagnostics are important as well as adequate strength intervention, to compensate for the unilateral deficit between the KF and KE muscle groups. The HY group in our study achieved an average result of 60.41 ± 9.87%. This result is related to the higher MS values of soccer players’ KFs when compared to the ASY and SY groups.

Our screened sample of soccer players (HY) showed comparable UR values (H:Q_DL_: 59.66 ± 8.81%, H:Q_NL_: 60.41 ± 9.87%) with those reported by Maly et al. [12] (H:Q_DL_: 58.95 ± 8.38%, H:Q_NL_: 56.58 ± 8.31%). Bonetti et al. [38] also presented comparable, slightly higher values for the dominant lower limb (H:Q_DL_ = 60.33 ± 18.06%) than non-dominant (H:Q_NL_ = 53.15 ± 10.84%). A significant difference (*p* < 0.05) was only found at an angular velocity of 60°·s^−1^. Hewett et al. [46] reviewed 22 studies involving 1455 men (non-athletes) and demonstrated a significant correlation between the H:Q ratio and angular velocity for isokinetic testing (higher velocity = higher H:Q ratio). Despite this, Magalhaes et al. [37] reported significant differences in the UR between the DL and NL in volleyball and soccer players at 90°·s^−1^. Rosene et al. [47] did not find any significant differences in the H:Q ratio, after examination of UR in male (soccer, volleyball) and female athletes (softball, soccer, basketball). This finding indicates a specific relationship between the peak torque and muscle contraction velocity of KFs and KEs when the hamstrings have a greater capacity to produce MS at higher angular velocities than the quadriceps. Since the thigh muscles of athletes usually work at higher movement velocities (kicking, jumping, swinging, throwing), it can be argued that isokinetic tests at higher angular velocities could have more informative value concerning the specific movements of the selected sports specialization, but on the other hand evaluation of maximum peak torque at slower speeds represents a starting point of relatively maximal MS ability [48]. Moreover, unilateral strength ratio at slow angular speed (60°·s^−1^) was reported as a strong predictor for a non-contact leg injury in National College American Association athletes [49].

Since individual playing positions in team sports have specific demands in terms of physical, technical, and tactical parameters according to results representation and comparison, players should be compared according to their positions (e.g., soccer: goalkeeper, fullback, center back, etc.).

We consider a smaller group sample and the absence of higher angular velocities evaluation (180 and 300°·s^−1^) as the main limitations of the study. Involvement of eccentric MS evaluation is also very important in sports, where maximum running speed interferes with rapid deceleration or change in directions. Another limitation of the study is gender indistinguishability. The presented results should be related to athletes undergoing puberty. Future research should consider incorporating peak height velocity in participant selection, widen the sample group for better generalizability of the results, and evaluate more types of sport, age, and performance level categories.

## 5. Conclusions

Present research supports that the specific performance and load required in various sports with predominantly symmetrical, asymmetrical, and hybrid types of locomotion affect the level of bilateral and unilateral lower limb MS asymmetry in young athletes. The groups that performed symmetric and hybrid sports achieved significantly lower values of BA in their lower limb muscles than those who played asymmetric sports. The hybrid sports group achieved significantly higher UR values in DL and NL. We can state that symmetrical movements that combine cyclic and acyclic movement patterns (which occurs frequently in hybrid sports) can bring a positive balance when it comes to loading both lower limbs equally, and the level of asymmetry could naturally decrease. The results showed large intra-individual differences even within individual groups (SY, ASY, and HY), which may be reduced by assigning players according to their playing positions in the future. Moreover, movements with a predominantly symmetrical character (walking, running, swimming, cycling) can be strongly influenced by an individual’s execution of the movement, fatigue, and speed. Therefore, we recommend regular monitoring of imbalances in the manifestation of BA and UR strength in young athletes of all sports. Long-term isokinetic monitoring at repeated periods can be used to analyze a player’s MS abilities as well as the effectiveness of the training process concerning elite sports. These results should be helpful for specialized and clinical workers including athletic trainers, strength and conditioning trainers, fitness coaches, or physiotherapists. With appropriate adjustment of individual plans, the identified strength deficits should be balanced due to lowering risk factors and injury prevention across all age categories.

## Figures and Tables

**Table 1 medicina-56-00683-t001:** Comparison of the selected muscle strength asymmetries and ratios between knee flexors and knee extensors in the monitored groups.

	Symmetric		Asymmetric		Hybrid		F_2,47_	*p*	$\eta_{p\;}^{2}$
	**x**	**SD**	**x**	**SD**	**x**	**SD**			
Q:Q (%)	5.42	3.02	9.40 ^a^	4.32	5.46	5.70	4.74	0.01	0.17
H:H (%)	8.30	7.79	11.74	7.41	6.97	7.01	2.08	0.14	0.08
H:Q_DL_ (%)	53.55	8.92	51.18	6.57	59.66 ^b^	8.81	5.21	0.01	0.18
H:Q_NL_ (%)	53.18	9.36	51.81	7.98	60.41 ^b^	9.87	4.40	0.02	0.16

x—mean; SD—standard deviation; Q:Q—quadriceps to quadriceps ratio; H:H—hamstring to hamstring ratio; H:Q_DL_—hamstring to quadriceps ratio for the dominant lower limb; H:Q_NL_—hamstring to quadriceps ratio for the non-dominant lower limb; ^a^—higher value than symmetric and hybrid, ^b^—higher value than symmetric and asymmetric.

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
