# Peer review of "Unilateral and Bilateral Strength Asymmetry among Young Elite Athletes of Various Sports"

_medicina, 2020, doi:10.3390/medicina56120683_

Round 1

Reviewer 1 Report

Dear Authors,

Thank you very much for your interesting manuscript.

Altogether it is well-written and worth being published in Medicina. However, before I can recommend its publication, please address the following issues:

Major issues:

  • You have assigned triathlon a symmetric MS character. Although I understand your basic (implicit) argumentation for that, it is imperative to see that athletes in such “symmetric” sports like triathlon or distance running still often exhibit substantial asymmetries. This was recently discussed in, e. g. https://doi.org/10.1016/j.orthtr.2019.06.001 and https://doi.org/10.1016/j.orthtr.2019.06.002. You should at least acknowledge that in your discussion.
  • You have measured solely at 60 °/s, while also 180 °/s is often a second standard angular speed for isokinetic testing. As you correctly state in line 222, the hamstrings often possess higher speed capabilities. So please discuss why you have not at least there measured also at 180 °/s.
  • It remains unclear to the reader how you determined the dominant limb. Please describe that procedure (testing, questionnaire etc.?)
  • You worked with adolescent subjects. Was your study carried out in accordance with the declaration of Helsinki? If so, please explicitly state it. If not, I have severe ethical concerns.

Minor issued:

  • In your introduction or discussion, you should at add and briefly discuss another recent work on exactly that subject in the context of your study: https://doi.org/10.1016/j.orthtr.2019.01.007
  • Line 117: The meaning of the term standard deviation is clear to the readership, so no need to explain it in parentheses.

Best regards

Your reviewer

Author Response

Dear reviewer,

thank you very much for very stimulating and helpful comments. We realized that the submitted work can be improved based on the assessment and incorporation of comments. We appreciate your time and effort in order to provide your revision with aim to improve the submitted manuscript. We believe that the new version has a higher quality and would be interesting for readers. We believe that the published research issue will find a large circle of readers and will also be cited in future articles. Once again, we would like to sincerely thank you for your time, effort, professional and factual comments.

Dear reviewer, please see the attachment where are all comments and changes mentioned. Also, have a look a new version of paper with  all correction, what has been made base on both reviewers.

Yours authors.

Reviewer 2 Report

General Comments

This study investigated unilateral and bilateral asymmetries in different types of sports. This topic has clinical relevance as asymmetries are often discussed as an injury risk factor. While the study attempts to answer an interesting question, the manuscript has several areas of needed improvement.

The introduction, particularly L49-80, is more of a literature review with results from previous studies simply listed without much context or explanation of relevance. The goal of the introduction should be to provide a brief review of the literature and explain the gap in literature. I believe the authors can shorten the literature review and provide more context as to the gap in the literature that exists. Additionally, more justification as to how the different sport types were selected would be helpful.

The methods are lacking detail about the participants and signal processing. How was it determined the participants were “elite” and how was specialization confirmed? Were they males and females? How was the peak torque determined? Was the signal filtered?

The presented results are confusing. The first sentence (L127-128) presents a main effect for sport, but it is unclear how this was determined. The authors stated in the methods that ANOVAs were used, so it is unclear how this result was determined. It is also noted in the methods that Shapiro-Wilk and Levene’s tests were performed; however, neither of these results are presented – the latter of which is relevant due to the standard deviations that are presented. There is also a great deal of redundancy of presenting the means and standard deviations (i.e., the results, the table, and the discussion). Removing some of the redundancy and including more context of the results would help the manuscript and the reader. For example, effect sizes are presented but never discussed.

The discussion needs to explain the findings more and not just repeat the results and list the results of the previous studies. I thought section 4.2 did a better job of explaining the findings than section 4.1. I would also suggest having a paragraph before section 4.1 that recaps the major findings of this study and then have the different sections that discuss bilateral and unilateral findings.

Overall, I would suggest removing many of the abbreviations. While abbreviations shorten the length of the manuscript, the ones used are uncommon and some of the abbreviations are only used a few times – this makes it hard to read as the reader has to stop and recall what the abbreviations mean.

Specific comments:

L32: It is unclear what is meant by “predominant character” – please explain/define.

L35-37: This sentence is important for setting up the context of the study, but no context is provided. I suggest expanding on why this information is important. I.e., what are the consequences of this asymmetries.

L38: It is unclear what “natural principles of motor learning” means. Is this referring to missing attainment of certain fundamental motor skills or something else? Please clarify.

L49-55: These sentences try to explain the different sport classifications, but many don’t relate to the current study. I suggest cleaning this up so the reader understands why the sports that were selected in this study are important to study.

L70-71: This sentence needs to be reworded. While the cited study reported a correlation between strength and change of direction performance, that does not mean that it clearly shows that increasing strength improves change of direction performance. There are other factors that can influence change of direction performance.

L72-74: This is an interesting finding and deserves more discussion. I suggest the authors bring this up in the discussion as it provides some important context related to the results.

L78: I recommend removing the p value from this sentence as it doesn’t add much to the sentence.

L93: It is unclear what “aerobics” entails. Please specify.

L105: I have not looked at the journal’s requirements, but do they request degrees or radians to be reported?

L107-108: This is unclear. Please clarify.

L153-161: This seems like it should be a separate paragraph as it is very specific, whereas the first few sentences of the paragraph are more general results.

L177-178: Check the wording of this sentence for clarity.

L188: It is unclear why the results are gender specific. Again, it is unclear in the methods who the participants were.

L217: Please define what is meant by “conventional” H:Q.

L223-226: The authors should justify why they chose to evaluate strength at 60 degrees per second.

L233-234: Was peak height velocity used in participant selection? If so, this needs to be explained in the methods.

Table: I do not think the ratio needs to have (%) since the reader should understand what a ratio is. However, I will let the editor decide if this should be included.

Author Response

Dear reviewer,

thank you very much for very stimulating and helpful comments. We realized that the submitted work can be improved based on the assessment and incorporation of comments. We appreciate your time and effort in order to provide your revision with aim to improve the submitted manuscript. We believe that the new version has a higher quality and would be interesting for readers. We believe that the published research issue will find a large circle of readers and will also be cited in future articles. Once again, we would like to sincerely thank you for your time, effort, professional and important comments.

Dear reviewer, please see the attachment where are all comments and changes mentioned. Also, have a look a new version with  all correction, what has been made base on both reviewers suggestions.

Yours authors.
